# A Closer Look at Domain Shift for Deep Learning in Histopathology

Karin Stacke[1,3], Gabriel Eilertsen[1,2], Jonas Unger[1,2], and Claes Lundström[1,2,3]

[1] Department of Science and Technology, Linköping University, Sweden
[2] Center for Medical Image Science and Visualization, Linköping University, Sweden
[3] Sectra AB, Sweden
{karin.stacke, gabriel.eilertsen, jonas.unger, claes.lundstrom}@liu.se

**Abstract.** Domain shift is a significant problem in histopathology. There can be large differences in data characteristics of whole-slide images between medical centers and scanners, making generalization of deep learning to unseen data difficult. To gain a better understanding of the problem, we present a study on convolutional neural networks trained for tumor classification of H&E stained whole-slide images. We analyze how augmentation and normalization strategies affect performance and learned representations, and what features a trained model respond to. Most centrally, we present a novel measure for evaluating the distance between domains in the context of the learned representation of a particular model. This measure can reveal how sensitive a model is to domain variations, and can be used to detect new data that a model will have problems generalizing to. The results show how learning is heavily influenced by the preparation of training data, and that the latent representation used to do classification is sensitive to changes in data distribution, especially when training without augmentation or normalization.

**Keywords:** histopathology · domain shift · cross-dataset generalization

## 1 Introduction

A fundamental part of machine learning is the problem of generalization, that is, how to make sure that a trained model performs well on unseen data. If the unseen data has different distribution, i.e. a *domain shift* exists, the problem is significantly more difficult [15,22] – even the smallest changes in the statistics as compared to the training data can cause a deep convolutional neural network (DNN) to fail completely [10].

In the case of digital pathology, domain shift challenges in whole-slide-imaging (WSI) are typically ascribed to color, brightness and contrast variations, caused by stain variations and scanner properties [24]. Diversifying the training data can alleviate the problem, e.g., by using samples from different medical centers. However, there are still no guarantees that the trained model will generalize to all situations faced by a real-world application.

Much research has been devoted to understand DNNs trained with natural images, for example what representations the model learns, how the choice of loss function and batch size affects the representation [7,6,9] and how this can be leveraged for transfer learning, novelty detection and detection of adversarial examples [6,17,14]. However, the relevance of these research results for histopathology is unclear, as the data characteristics substantially differ from natural images. As Raghu et al. observed in [16], models trained on medical image datasets did not learn Gabor filters, which were the case when trained on ImageNet [4]. The same is true for models trained with histopathological data, as demonstrated in Figure 1. While Gabor filters are known to be a common denominator between convolutional neural networks (CNNs) trained for different tasks and on different data, this counter-example motivates us to look closer at how CNN optimization is shaped by WSI data, and what the implications are for generalization to unseen domains.

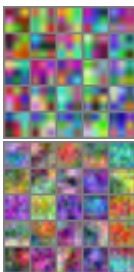

**Fig. 1:** Visualizations of *non*-Gabor-like filters from the first convolutional layer for Simple CNN (top) and Mini-GoogLeNet (bottom) architectures, trained with histopathological data. See Section 2 for model definitions.

The problem of domain shift in pathological images from different medical centers and different scanners has been discussed in previous work [20,11,2,1,3,21], but not with the purpose of getting a deeper understanding of the underlying elements of the problem. In this paper, we take a closer look at how different strategies for increasing generalization affects not only performance, but also the representations learned by a CNN trained on histopathology data. To this end, we look at the problem from three different perspectives, which constitute the main contributions of this paper:

1. We analyze the test accuracy of CNNs trained for tumor classification on H&E stained images: how it changes with different models, test data and augmentation strategies.
2. We optimize for the input images which maximally activate different convolutional filters of a trained model, to get an indication on what particular features a model respond to.
3. We use the filter activations of a trained model to formulate a novel representation shift metric that captures differences in feature representation caused by test data from a different distribution than the training data.

The experiments show that a tumor classifier respond to very different features depending on how training data is processed. We are also able to demonstrate a correlation between the representation shift and the drop in classification accuracy on images from a new domain. Together, the experiments help us shape a better understanding on what a tumor classifier has learned and how sensitive its representation is to changes in data distribution. We believe that this understanding is crucial for future development of reliable deep models that are robust to domain variations and thus possible to use in a real-world clinical pipeline.

## 2 Experimental setup and methods

Below, we first describe the data and models used, followed by the setup of the three experiments of our study. Finally, the details of our proposed domain shift quantification are given.

### 2.1 Data

We use the CAMELYON17 dataset [12], consisting of H&E stained lymph node whole-slide images, captured with three different scanners, (henceforth denoted *Scanner 1*, *Scanner 2* and *Scanner 3*) collected from three, one and one medical centers, respectively. Three types of data transformations were included (Fig. 2), in order to measure how well traditional and state-of-the-art methods can overcome the domain shift between scanners:

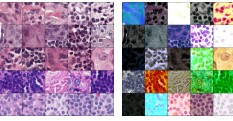

Original    Color augm.

Stain norm.CycleGAN

**Fig. 2:** Five example images for each center (the three top rows are scanned with Scanner 1, the forth row with Scanner 2 and the last row with Scanner 3).

**Color and intensity augmentations** (henceforth shortened *color augmentation*) aim to increase image diversity in order to make the model more robust to color variations. The augmentation was performed randomly on each channel of the HSV color space. The amount of augmentation was chosen for best generalization, which meant extreme and unnatural color variations. However, this was necessary in order to force a model to focus on other features than color.

**Stain normalization** refers to first decomposing the image by stain colors (stain separation), and then normalizing it based on a target distribution. A wide range of stain normalization techniques has been presented. For this study, the method presented in [23] was used. Each patch was normalized in relation to a reference patch, taken as a representative patch (in terms of structure and color) from one of the medical centers (Canisius-Wilhelmina Hospital, CWZ).

**CycleGAN** [25] is a method for image-to-image translation, with the goal of translating an image using a mapping $G$, from source domain $X$ to target domain $Y$. The goal is to learn this mapping $G$, such that the distribution of $G(X)$ is indistinguishable from $Y$. In order to achieve this, the inverse mapping $F : Y \rightarrow X$ is introduced to force $F(G(X)) \approx X$. Using this approach, features important to the target domain is transferred to $X$, but unimportant ones are unchanged. In our work, we use this approach to transfer images from one medical center to another. The class label of the image is left unchanged, only the features that are different between the centers are changed. Images from all centers where transferred to medical center CWZ.

### 2.2 Model architecture

Two model architectures were evaluated. One simple CNN architecture consisting of three convolutional layers and two fully connected layers (with dropout, no

batch normalization), and a small version of GoogLeNet architecture [18] (only using the output from the first auxiliary classifier, henceforth denoted *Mini-GoogLeNet*). The motivations for choice in architecture is the small training dataset requiring a reduced size in model capacity to prevent overfitting, and as the goal of this experiment is not to achieve state-of-the-art results, simple and small models are to be preferred. We also tested larger and more high-capacity models, such as ResNet [8] and Inception-v3 [19]. However, the performance did not improve, and in most cases overfitting was a significant problem. All models were trained with geometric augmentations (random flip, rotation, and crop), with a patch size of 96x96px, and a resolution of 0.25 microns per pixel. The datasets were separated at slide level, making sure no patches from the same slide were in both training and test sets. All results shown are the average of three training sessions.

### 2.3   Experiments

The three domain shift analysis experiments were designed as follows.

**Cross-dataset generalization for tumor classification**   was evaluated by training CNN models as *tumor classifiers*, detecting tumor vs. non-tumor patches. The training data consisted of slides scanned with *Scanner 1*. For test, both same-scanner data as well as unseen data from the two other scanners was used, with results reported as patch classification accuracy.

**Feature visualization** of learned model filters was employed to give a visual representation of what image features the model respond to. Using a gradient *ascent* method [5,13] we start from a noise image and iteratively optimize towards the input image which maximally activates a filter in the last convolution layer.

**Activation difference** for filters responding to data from different domains was the third aspect in focus. To study this, we developed a quantitative metric, *representation shift*, described in the next section. The metric was applied to the learned models of the cross-dataset generalization experiment.

### 2.4   Representation shift metric

To quantify domain shift in the context of a CNN, we are not only interested in the statistical differences between images from different domains, but also in how the images are processed by the CNN. For this purpose, we propose the *representation shift* metric. This measures the differences in distributions of the learned feature representation, comparing the training set to a dataset from a different domain. We denote by $p_{r,c_i}$ the continuous distribution of mean activations from the convolutional filter $c_i$ in the last convolutional layer $\{c_1, ..., c_L\}$, computed using input data $X_r = \{x_{r1}, ..., x_{rn}\}$. A second dataset, $X_\theta = \{x_{\theta 1}, ..., x_{\theta m}\}$ similarly generates $p_{\theta,c_i}$. If $\pi(p_{r,c_i}, p_{\theta,c_i})$ are the joined distributions with margins $p_{r,c_i}$ and $p_{\theta,c_i}$, the Wasserstein distance between the distributions is given by:

$$W(p_{r,c_i}, p_{\theta,c_i}) = \inf_{\pi \in \Gamma(p_{r,c_i}, p_{\theta,c_i})} \int_{RxR} |(x-y)| d\pi(x,y). \tag{1}$$

We define the *representation shift* $R$ as the mean distance between the distributions over all filters $c_i$,

$$R(p_r, p_\theta) = \frac{1}{L} \sum_{i=1}^{L} W(p_{r,c_i}, p_{\theta,c_i}).$$ (2)

If $X_r$ and $X_\theta$ are statistically similar in the image domain, or if the CNN maps the datasets to similar representations, filter responses should be similar, and $R(p_r, p_\theta)$ small. We can then expect a similar classification accuracy of the datasets. If $R(p_r, p_\theta)$ is large, then filter responses has changed between the datasets, resulting in a higher risk of generalization error. The method does not aim to do out-of-distribution or novelty detection for individual images (as e.g. [14,17]), but aims to indicate an elevated risk of decreased classification accuracy for a dataset that is assumed to be similar to the original data. As the metric does not require annotated data, it can serve as a simple initial test to evaluate if new data (e.g. from a new scanner) is handled well by an already trained model, i.e. if the learned feature representation applies to the new data. Also, as the metric is tightly connected to the specific model used, we expect that different models and training strategies will results in very different robustness to changes in statistics (Fig. 4).

## 3   Results

### 3.1   Cross-dataset Generalization

Table 1 shows validation accuracies when training a model on data from one scanner, and evaluating it on another. Both model architectures suffer from poor generalization when no augmentation is used, with significant drop in mean accuracy (21.7 percentage points (p.p.) mean drop from same-scanner data to unseen data). Adding color augmentation results in better generalization across the datasets (4.75 p.p. drop), with similar performances on both unseen datasets. Stain normalization performed adequate on one of the unseen datasets (Scanner 2), but had a quite large drop for the Scanner 3 data (mean of 9.2 p.p. drop). CycleGAN gave best performance on same-scanner data, but these high values in accuracy did not transfer to the other data sets (11.45 p.p drop).

### 3.2   Feature visualization

Images visualizing features that give maximal activation of filters in the last convolutional layer are shown in Fig. 3. Note that color reproduction may not

**Table 1:** Mean accuracies (%), training a tumor classifier on Scanner 1, testing it on same-scanner data and other data, using two different model architectures.

| Train \ Test | *Simple CNN* | | | *Mini-GoogLeNet* | | |
|---|---|---|---|---|---|---|
| | Scanner 1 | Scanner 2 | Scanner 3 | Scanner 1 | Scanner 2 | Scanner 3 |
| Orig. data | $88.0 \pm 3.4$ | $51.2 \pm 0.7$ | $78.4 \pm 1.7$ | $87.2 \pm 2.8$ | $60.0 \pm 14.4$ | $74.3 \pm 9.6$ |
| Color aug. | $87.2 \pm 4.4$ | $80.2 \pm 1.7$ | $\mathbf{82.3 \pm 0.8}$ | $87.3 \pm 4.5$ | $\mathbf{84.5 \pm 1.2}$ | $\mathbf{82.8 \pm 5.8}$ |
| Stain norm. | $89.4 \pm 3.0$ | $\mathbf{84.1 \pm 2.0}$ | $77.6 \pm 4.6$ | $88.1 \pm 1.5$ | $80.5 \pm 4.1$ | $77.9 \pm 6.9$ |
| CycleGAN | $\mathbf{94.5 \pm 0.2}$ | $83.2 \pm 5.3$ | $80.9 \pm 3.1$ | $\mathbf{91.0 \pm 1.8}$ | $83.4 \pm 3.1$ | $78.2 \pm 2.2$ |

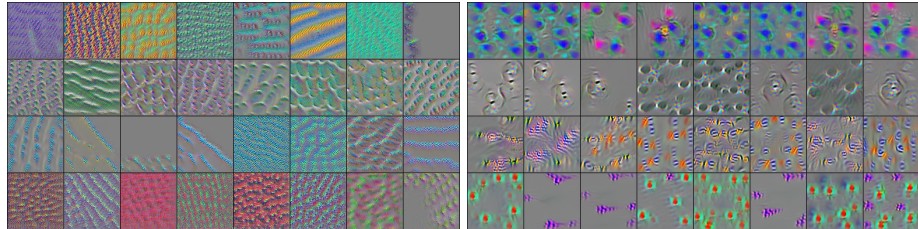

**Fig. 3:** Each row shows example images that maximally activates different filters in models Simple CNN (left) and Mini-GoogLeNet (right), trained with (from top to bottom): original data, color aug., stain norm., and CycleGAN. Best viewed digitally.

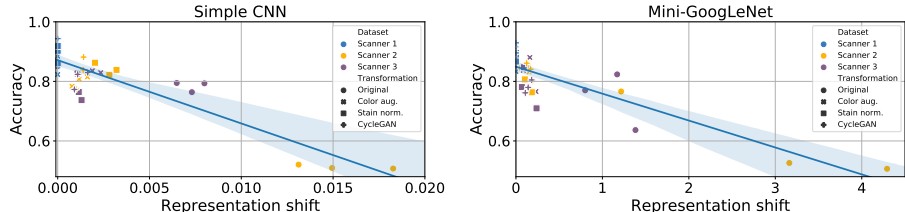

**Fig. 4:** Relation between patch level accuracy (see Table 1) and representation shift (Equation 2) for Simple CNN (left) and Mini-GoogLeNet (right) with and without data transformation. A regression line shows the correlation between the variables.

be completely accurate, since no regularization on maximal pixel values is done during the optimization process. However, the relative color differences between the models can be compared, showing that the Mini-GoogLeNet model trained with color augmentation completely ignores the color components in the images. This is in strong contrast with the model trained on original data, which seem to react to one or two colors only. Thus, it is not surprising that the generalization to another domain is poor – if the representation is highly dependent on color information, we can expect that small differences in color distribution will affect the performance negatively. Since differences in color is one of the most evident visual differences between images from different scanners, this is indeed the case.

Images from Mini-GoogLeNet reveal that the models have learned cell-like circular structures, and trained with color augmentation or stain normalization show even stronger resemblances with nucleus structures and cell membranes. The Simple CNN architecture seems to find more low-level structures, which might indicate that the model has not learned the expected features which separate tumor cells from non-tumorous tissue. Both stain normalization and Cycle-GAN transformation unifies the color variations in the dataset, which is visible by a more homogeneous color scale in these images.

### 3.3   Representation shift

Fig. 4 shows the representation shift in relation to model patch level accuracy. Models trained with original data show large representation shifts, which is to be

expected as no measures where taken to handle the domain shift. The representation shift is largely reduced with any of the data transformation techniques. There is not a one-to-one mapping between the representation shift and classification accuracy, which is expected as the final accuracy depends on the final fully connected layers. However there is a clear negative correlation between representation shift and classification accuracy.

## 4  Discussion

This study is an initial attempt to analyze how histopathological data shapes DNNs differently than natural images do. Although the presented techniques can be straight-forwardly used for detecting model vulnerability, we also believe that such information can be used to increase our knowledge about deep learning in histopathology.

Using feature visualization (Fig. 3), we see that different preparations of training data results in very different latent representations. Even if this way of visualizing the learned features only give us parts of the truth, it is an important insight into the "black box" of CNNs in histopathology. We see that these different representations are more or less sensitive to changes in input distsribution (Table 1). For example stain normalization or CycleGAN may unify the data and reduce much of the variance, which help the cross-dataset generalization, but does not necessarily create robust models.

The results also show how the representation shift metric can give a valuable contribution for domain shift analysis on a new dataset (Fig. 4), without requiring annotations on the new data. While it is still a coarse measure, we argue that it is useful tool for a first investigation of the magnitude of a domain shift.

In future work, we will further investigate the correlation between activations and domain shift. In particular, we aim to further explore representation shift metrics as a tool for precise domain shift analysis, and how this is affected by different models, datasets, and domain adaptation techniques.

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
