# OpenReview forum: "A closer look at domain shift for deep learning in histopathology"
_MICCAI.org/2019/Workshop/COMPAY — COMPAY 2019_

### Official Review · AnonReviewer3 · 2019-08-11
**Good discussion paper, but lacks novelty**

**Rating:** 5
**Confidence:** 3

**Review:**

Summary:

This paper explores domain shift within histology images due to variation in images between different centres and due to variation in the fixation and staining procedure. The difference in distribution of images between the training and testing sets has a direct impact on the generalisability of models.

The contributions of the paper are:
- Analysis the test accuracy of CNNs trained for tumour classification on H&E stained images: how it changes with different models, test data and augmentation strategies.
- We optimise for the input images which maximally activate different convolutional filters of a trained model, to get an indication on what particular features a model respond to.
- We use the filter activations of a trained model to formulate a novel representation shift metric that captures differences in feature representation caused by test data from a different distribution than the training data.

The first contribution has already been studied within computational pathology and therefore although it is an important component of the paper,  I would not consider it to be a major contribution. Also, I am not completely convinced that the filter visualisations can be used in a conclusive manner. The interpretation of these visualisations may be subjective. Therefore, I feel that the 3rd contribution is the main contribution of the paper and it should be the main focus. However, I have a couple of concerns:

First, it would be good for you to comment why you assumed a linear relationship between the accuracy and the representation shift.
It seems that the introduction of scanner 3 has led to this assumption of the linear relationship. However, if a different scanner instead of scanner 3 was introduced; would the same relationship be observed?

Overall, despite this discussion being interesting, I am not convinced that this work is strong enough to be published. A more interesting approach would be how to use this idea of representation shift to increase the performance of the model.

---

### Official Review · AnonReviewer2 · 2019-08-14
**No title**

**Rating:** 5
**Confidence:** 3

**Review:**

This paper presents an exploratory study regarding domain shift challenges for analyzing histology images. The study is performed on a publically available dataset (Camelyon17).
Tow model architectures (Simple CNN and Mini-GoogLeNet) were selected for the evaluation of patch-based classification with two classes tumor and non-tumor. The contributions of this work are summarised as follows 1) how augmentation strategies impact model performance, 2) visually analyze model activation, and 3) representation shift metric to quantify differences in distributions of two datasets. The paper is well structured, with a clear motivation to the problem

Below are some comments

- Some important details regarding the reported results are missing, which makes it difficult for readers to reproduce this work.  For example, the number of training, validation and the testing split is missing for each scanner, other details are batch size, learning rate, number of epochs, HSV intensities for color augmentations, etc
- The motivation for the selection of small models (less deep) over state-of-the-art models should be explained properly. Camelyon17 is considered as a reasonably good dataset so it is not clear how overfitting was the bottleneck in using relatively deep models
- A literature review or theoretical insights regarding the representation shift metric is missing. The motive of using Wasserstein distance is not clear.
- In Fig 4, it should be explained why there are 3 values for each experiment
- Input RGB images in Fig 3 would help in understanding activations
- If possible, the resolution of Fig 1 and 2 should be increased

---

### Official Review · AnonReviewer4 · 2019-08-15

**Rating:** 6
**Confidence:** 3

**Review:**

This paper tries to overcome the issue of domain shift in histopathological images due to scanner variations. The authors test the generalization capabilities of two CNN’s when no-augmentation, stain normalization, color augmentations, or cyclegans are used. To test the differences, they apply the techniques on a tumor vs. non-tumor classification task. They split the Camelyon17 dataset per scanner in a train and test set, all models have been trained on patches of 96x96px with 0.24um/px extracted from slides scanned by scanner 1.  The authors use two small models, a five-layer deep CNN and a small version of GoogleNet. They evaluate the differences between the models based on accuracy, filter visualization, and a proposed new metric ‘representation shift’. The ‘representation shift’ is the major contribution of the paper.

Pro’s
-	Clear problem statement and sound.
-	The authors proposed a new metric to quantify if a new dataset would give similar results from a CNN compared to the dataset it was trained on. It would be interesting to see how this could be used in an end-to-end training method and compared to different out-of-distribution methods.
Con’s
-	The comparison between the different augmentation/normalization methods has already been done by other studies. The literature discussion can be extended and it would be good to compare the results with the different papers.
-	The authors should explain the dataset in more detail. It is unclear how many of the 1000 slides of the Camelyon dataset are used and how many patches are extracted.  Also, the field of view of the models might be very limited by the choice to extract patches of 96x96 at pixel resolution 0.24 might.
-	It is unclear how is the cycle-gan trained? What models are used? Is it trained with slides from all scanners?
-	The authors only take into account standard color variations. However, stain variation is more than only color variation. It would be good to include some additional augmentation methods i.e. blurring, elastic deformation, etc.
-	It would be interesting to see how well the networks would perform when a network was trained on slides from two scanners. Also, it would be interesting to compare the results to other slides from, i.e. TCGA or another challenge.
-	Some figures are difficult to read when printed.
-	It is not easy to compare the models in figure 4 because the scales are very different. Also 95% confidence intervals should be included when correlation is calculated.
-	The visualization of the filters is difficult to interpret and subjective.

To conclude, the authors could consider to exclude the visualization part and extend the introduction, data and materials, and results.

---

### Decision · Program_Chairs · 2019-08-20

Accept